# UV-B Radiation Induced the Changes in the Amount of Amino Acids, Phenolics and Aroma Compounds in *Vitis vinifera* cv. Pinot Noir Berry under Field Conditions

**DOI:** 10.3390/foods12122350

**Published:** 2023-06-12

**Authors:** Meng Sun, Brian Jordan, Glen Creasy, Yi-Fan Zhu

**Affiliations:** 1College of Food Science and Technology, Yunnan Agricultural University, Kunming 650201, China; 2Centre for Viticulture and Oenology, Faculty of Agriculture and Life Sciences, Lincoln University, Christchurch 7647, New Zealand; sm183495665@163.com (M.S.); brian.jordan@lincoln.ac.nz (B.J.); creasyg@gmail.com (G.C.); 3Institute of Pomology, Jiangsu Academy of Agricultural Sciences/Jiangsu Key Laboratory for Horticultural Crop Genetic Improvement, No. 50 Zhongling Street, Nanjing 210014, China; 4SCEA Terre des 2 Sources, La Plaine, 34190 Montoulieu, France; 5University Engineering Research Center for Grape & Wine of Yunan Province, Yunnan Agricultural University, Kunming 650201, China

**Keywords:** UV-B radiation, amino acids, phenolic compounds, aroma compounds, *Vitis vinifera* ‘Pinot noir’

## Abstract

High UV-B radiation can challenge Pinot noir growth in the wine-making region of the Southern Hemisphere. The aim of this work was to determine UV-B effects on amino acids, phenolic composition and aroma compounds of Pinot noir fruit. Sunlight exposure with or without UV-B did not affect fruit production capacity, °Brix and total amino acids in the vineyard over the two years. This research reported increased contents of skin anthocyanin and skin total phenolics in berry skins under UV-B. The research showed that there were no changes in C6 compounds. Some monoterpenes concentrations were decreased by UV-B. The information also indicated how important leaf canopy management was for vineyard management. Therefore, UV radiation potentially affected fruit ripeness and crop load, and even stimulated the accumulation of phenolic compounds that may affect Pinot noir quality. This research reported that canopy management (UV-B exposure) may be a good way for vineyard management to increase the accumulation of anthocyanins and tannins in berry skins.

## 1. Introduction 

Grapevine berry quality for wine making was highly dependent on the environment and climate factors [1]. Ultraviolet-B (UV-B) radiation strongly influenced the compositions of grape berry and resulting wine [2]. High UV-B has been traditionally regarded as an abiotic stress for plants, which can lead to diverse physiological damage, such as oxidative damage on DNA, pigment degradation and yield reduction [3]. However, many studies indicated that increased UV-B radiation improved the quality of grape berry and resulting wine, especially on the phenolic and aroma compounds [2,4,5,6]. Moreover, high-fluence UV-B stimulated non-specific signal transduction by the upregulation of genes through jasmonic acid (JA), salicylic acid (SA) and the ethylene pathway [7,8]. Therefore, the different fluence of UV-B induced the different signaling transduction pathways intermediating secondary metabolites in plants. 

Phenolic compounds are a class of the most important plant secondary metabolites and significantly contribute to grape and wine quality [9]. They are made up of six carbon atoms, with one or more hydroxyl groups or derivatives of this basic structure. The hydroxyl groups and unsaturated double bonds lead to phenolic oxidation [10]. Phenolics were divided into non-flavonoids and flavonoids in grapes. Flavonoids were present in high concentrations in grapes [9,11]. These were C6-C3-C6 polyphenolic compounds that had a heterocyclic C ring (a three-carbon chain) between two hydroxylated benzene rings. According to the oxidation of the C ring, flavonoids in grapes were divided into three major classes: flavonols, flavan-3-ols and anthocyanins [9]. The phenolic composition of plants, especially flavonoids, can be dramatically increased due to the regulation of the flavonoid biosynthetic genes by low -fluence UV-B radiation [6,12,13,14]. Flavonol concentration in grape skin was significantly affected by the environmental elements, particularly UV radiation [4,5]. 

Grape volatile compounds played a critical role in the quality of grape berry and wine. The aroma compounds of grapevine and their precursors were established by secondary metabolites during the second growth phase [15]. Some odor compounds were stored in grapes as water-soluble glycosides or combined with amino acids [16]. In these forms, the compounds cannot be detected by smell until glycosidases and peptidases release the volatile compounds from the water-soluble forms to the wine aroma [17]. UV-B also affected the synthesis of carotenoids, which are tetraterpenoids (one class of isoprenoid composition) and precursors to norisoprenoids compounds [6]. Norisoprenoids (such as β-damascenone and β-ionon) were the critical compounds constituting grape varietal aroma [12]. Carotenoids, as photosynthetic pigments, were involved in photoprotection and react with free radicals, such as ROS and superoxide [13,14]. Moreover, UV-B caused an increase in carotenes and xanthophylls, which were associated with non-photochemical quenching, to protect photosynthetic machinery against over excitation and ROS [14,18], but carotenoid biosynthesis was impeded by chronic UV-B [19].

Amino acids were the subunits for proteins and enzymes, and were also nitrogen and energy sources for yeast and bacterial metabolism [20,21]. Amino acids in grapes were precursors of aromatic compounds being metabolized to higher alcohols, aldehydes, organic acids, phenols and lactones [22,23]. In grape berry, glutamine (Gln) was converted into other amino acids by aminotransferases, such as proline (Pro) and arginine (Arg), which occupied the greatest percentages of total amino acids in grapes [24]. However, there was little known about sunlight regulation of amino acids in grapevines, particularly UV-B radiation regulation.

Based on this knowledge, UV-B was an environmental factor for vineyard management in the grape-growing region. This study contributes to the understanding of UV-B radiation exposure and the alteration of the chemical composition of fruit in *Vitis vinifera* L. var. Pinot noir. Pinot noir vines in the vineyard were subjected to different UV radiation initiated from veraison, and the effect on the composition of quality-related compounds in fruit was measured. Finally, this research can provide important information for the wine industry, with an investigation into how grapes were affected by the naturally high levels of UV radiation.

## 2. Materials and Method 

### 2.1. Sites and Materials

In the field trial, the vineyard was located at Lincoln University, Canterbury, New Zealand (43°39′ S, 172°28′ E), a cool climate area. The Pinot noir vines (clone 777 on 3309 rootstock) were planted in 1999 in a north–south row orientation with 1.2 m between vines and 2.5 m between rows. Vines were trained with two bilaterally opposed canes in a vertical shoot positioned system (VSP). All the grapes were harvested by hand in April 2016 and 2017. This trial was from veraison to harvest (from February to April 2016 and from February to April 2017). 

### 2.2. Treatments

The trial design was three UV-B treatments in eight replicated blocks (control treatment, leaf removal treatment, and polyethylene terephthalate screen (PETG) treatment, each treatment applied in four blocks in each row) (Figure 1). All vines were randomly selected in the vineyard, and buffer vines were used to avoid the impact of UV-B on each vine.

Twelve grapevines across two rows were divided into three groups (each group including four replications). UV-B exclusion was achieved using the method of Gregan et al. Gregan, Wargent [25] and Liu, Gregan [4]. A-frame-mounted transparent screens (240 cm × 60 cm) containing UV-B exclusion materials were placed over individual vines to cover the fruiting zone of the test vine and buffer on either side. In each group of vines, the following treatments were applied from veraison to harvest (Figure 1): (i) control treatment (SC): leaves around the fruiting zone were removed and clusters were covered by shade cloth, which could provide canopy-like shelter without changing leaf areas (Ultra-Pro 70% shadecloth, Cosio Industries Ltd., Christchurch, New Zealand); (ii) leaf removal treatment (LR): all leaves and lateral shoots were removed in the bunch zone leaving clusters fully exposed; (iii) PETG (glycol-modified polyethylene terephthalate, Mulford Plastics, Christchurch, New Zealand): all leaves and lateral shoots were removed in the bunch zone and clusters were covered by a PETG screen. In all treatments, leaves in the fruiting zone were removed to maintain the same leaf areas across treatments. 

### 2.3. Sample Collection

For the UV trials in the Lincoln University Research Vineyard, whole berries were collected at five stages of development: 1 week after veraison (13 weeks post-bud burst); 2 weeks after veraison; 3 weeks after veraison; 4 weeks after veraison; 5 weeks after veraison (harvest, 17 weeks post-bud burst). At each time point, samples from four replicates were randomly collected from the control treatment and UV-B treatment and immediately stored in a walk-in freezer (−20 °C). Ten berries from each replicate were randomly collected from different sides of clusters for the analysis of phenolic composition. At harvest, sample collection of 10 berries, 20 berries and 40 berries per replicate from each treatment were taken for the analysis of berry parameters, amino acids and volatile composition, respectively. 

### 2.4. Measurement of Chemical Analysis in Grapes

#### 2.4.1. Total Soluble Solids (TSS), pH and Titratable Acidity (TA) in Grape Juice 

Frozen berries collected at five stages of development were defrosted under room temperature, then fruit TSS, TA and the pH of the grape juice were measured using the method of Iland, Ewart [26]:

A small volume (2 mL) of juice from the berries was used to measure °Brix using a digital refractometer (PAL-1 ATAGO, Tokyo, Japan). Grape juice pH was measured by a Suntex pH/mV/temperature meter (SP-701; Suntex, Taiwan) with a Eutech Instruments probe (EC 620133; Eutech Instruments Pte Ltd., Singapore). Before the analyses, two standard buffer solutions of pH 4.0 and 7.0 were used to calibrate the pH meter. 

Titratable acidity (TA) was determined by titration to pH 8.2 using 0.1 mol/L NaOH (LabServ, 97% min; Biolab (Sydney, Australia) Ltd.). TA was measured on 10 mL of juice for the samples. NaOH (0.1 mol/L) was carefully added into the grape juice under constant stirring using a burette and the volume (mL) used for titration until pH 8.2 was recorded and used for calculations:Titratable acidity(gLasH2T)=75×molarity of NaOH×Titrevalue(mL)÷Volume of juice(mL)

#### 2.4.2. Amino Acids Analysis

The frozen berries were ground with liquid nitrogen using a pestle and mortar, transferred into 10 mL tubes and centrifuged for 5 min at 6000 r/min. The supernatant was diluted with deionized water (1:4) in a new tube. The grape juice samples were filtered through a 0.45 μmol/L nylon 1 mL syringe filter into an HPLC glass vial. An internal standard, γ-aminobutyric acid (γ-GABA), was added to a final concentration of 100 μmol/L and capped tightly. For inline-derivatization of the primary amino acids, ơ-phthaldialdehyde was used as a fluorescence derivative; iodoacetic acid/mercaptopropionic acid was used to increase cysteine sensitivity, and 9-fluorenylmethyl chloroformate was a fluorescence derivative for proline [25]. 

We used the method of chromatography followed by Gregan, Wargent [25]. The samples were injected into an HPLC system (Hewlett-Packard Agilent 1100 series, Waldbronn, Germany) with a 250 × 4.6 mm, 5 μm Prodigy C18 column (Phenomenex, Milford, MA, USA). Data were analyzed using the Chemstation (Agilent) chromatography data system. The mobile phase consisted of two solvents: solvent A (0.01 mol/L Na_2_HPO_4_ with 0.8% tetrahydrofuran, adjusted to pH 7.5 with H_3_PO_4_) and solvent B (20% solvent A, 40% methanol, 40% acetonitrile). The gradient program was 0 min, 0% B; 14 min, 40% B; 22 min, 55% B; 27 min, 100% B; 35 min, 100% B; 36 min, 0% B, with a flow rate of 1 mL/min. For detection, a fluorescence detector was used with an excitation at 335 nm and emission at 440 nm. At 25 min, the detector was switched to a second channel (excitation at 260 nm and emission at 315 nm) to detect proline. Amino acids were identified by their retention time and their semi-concentrations were calculated in parallel to calibrate the internal amino acid standard (γ-GABA, 100 μmol/L).

#### 2.4.3. Berry Phenolic Compounds Analysis

Grape phenolic substances were extracted and analyzed following the procedures described by Iland, Ewart [26] and Bonada, Jeffery [27]. Skins were separated from the pulp of berries using tweezers and scalpels. Skins were extracted in 20 mL conical flasks containing 10 mL of 50% *v*/*v* ethanol for 30 min. Flasks were filled with nitrogen before being sealed to prevent oxidation. The flasks were then placed into a warm bath shaker (100 rpm, 22 °C) for 24 h in the dark. The extracts were pooled into centrifuge tubes and centrifuged for 5 min at 6000 r/min. 

Measurements at 280 nm were carried out on a Shimadzu UV-1800 Spectrophotometer (Shimadzu Corporation, Kyoto, Japan), using UV semi-micro 1.5 mL disposable cuvettes. The results were reported on the content of total phenolic substances per berry:Skin phenolic substances(auberry)=A280 nm×DF×EV×0.001

Measurements at 520 nm were carried out on a Shimadzu UV-1800 Spectrophotometer (Shimadzu Corporation, Kyoto, Japan), using 1.5 mL disposable cuvettes. The results were reported in per milligram of malvidin-3-glucose equivalents per berry:Skin anthocyanins(mgberry)=A520 nm÷500×DF×EV
where DF was the dilution factor of the extract in 1 mol/L HCl and EV was the extracted volume after maceration with 50% ethanol. The value of 500 was based on a previous report that estimated the extinction coefficient of malvidin-3-glucose in g/100 mL of solution. 

Skin and seed tannins analysis: Before the analyses, epicatechin was used as a standard for each batch of samples. Aqueous -epicatechin (Sigma-Aldrich E1753) solutions (10, 25, 50, 75, 100, 150 mg/L epicatechin) were used to establish a standard curve for reporting tannin absorbance (y = 0.0143x + 0.138, R^2^ = 0.9754). All A280 (tannin) values were reported in mg/L or g/L epicatechin equivalents of the original sample.

Skin and seed tannins were measured by the methylcellulose precipitation (MCP) tannin assay using the 1 mL assay in 1.5 mL disposable tubes [28]. 0.3 mL of methylcellulose solution (0.04% *w*/*v*, 1500 cP viscosity at 2%, M-0387, Sigma-Aldrich, CA, USA) was added to 0.1 mL of skin or seed extract solution. After 3 min, 0.2 mL of saturated ammonium solution (Sigma-Aldrich, Auckland, New Zealand) was added into the mixed solution and made up to 1 mL with deionised water. The solution was mixed well, left to stand for 10 min, then centrifuged at 8936 g for 5 min (Table 1). Measurements at 280 nm were carried out on a Shimadzu UV-1800 Spectrophotometer (Shimadzu Corporation, Kyoto, Japan), using UV (methacrylate) semi-micro 1.5 mL disposable cuvettes. For the control samples, 0.2 mL of saturated ammonium solution was added to 0.1 mL of the extract solutions and made up to final volume 1 mL with deionized water (Table 1). The solution was mixed well, stood for 10 min, then centrifuged at 8936 g for 5 min and measured at 280 nm.

A280 of the tannin in the sample solutions can be calculated by subtracting A280 (treatment) from A280 (control). Epicatechin solution was calculated by epicatechin equivalent calibration curve, ranging from 0 mg/L to 150 mg/L. The dilution factor for the skin or seed extract solutions was 10. The conversion to mg/g and mg/berry in seeds and skins from mg/L in the extract is shown below:$$Tannin\;content\;of\;seeds\;or\;skins\left(\mathrm{mg}/\mathrm{berry}\right)=\frac{\left[Tannin\right]e\times Ve}{No.}$$
[Tannin]e = tannins concentration in extraction (mg/L epicatechin eq.),Ve = final volume of extraction (L), andNo. = initial number of berry samples.

#### 2.4.4. Volatile Compounds Analysis

The analysis of six C6 and monoterpene volatile compounds in Pinot noir juice (Table 2) was determined using an automated HS-SPME GCMS (Headspace Solid-Phase Micro-Extraction Gas Chromatograph Mass Spectrometry) technique, based on the work of Canuti, Conversano [29], Dennis, Keyzers [30], Fan, Xu [31], Fang and Qian [32] and Yuan and Qian [33]. This adapted method utilised three synthetic deuterated internal standards; hexanal (d12) and hexyl (d13) alcohol and linalool (d3) all obtained from CDN isotopes (Sci Vac Pty Ltd., Montreal, QC, Canada). Eleven non-deuterated standards were used to generate standard curves for quantitative analysis. *E*-2-Hexenal was obtained from Acros Organics while all other non-deuterated standards were obtained from commercial supplier Sigma-Aldrich.

### 2.5. Statistical Analyses

Statistical analysis was undertaken using IBM SPSS Statistics 22. The data were subjected to an independent-samples T-test and two/three-factor analyses (ANOVA) to partition the variance into the main effects (UV-B and water deficit; UV-B, water deficit and time) and the interaction among them. In the case of significant interactions among factors, treatments were compared using the least significant difference (LSD) at the 5% level (*p* < 0.05).

## 3. Results

### 3.1. Effects of UV-B Exposure/Exclusion on the Physiology in Pinot Noir Vines

In the 2015/2016 and 2016/2017 vintages, the UV-B exposure, UV-B exclusion and shading results and climatic data [34] are presented in Table 3 and Table 4, respectively. Compared with shading (SC), UV-B exposure (LR) and UV-B exclusion (PETG) caused increased in °Brix at harvest, but there were no effects on TA and pH. In the 2016–2017 season, there was a significant reduction in TA in the UV-B exposure (10.4 g/L) and UV-B exclusion (10.5 g/L) treatments in comparison with the shading treatments. According to pruning weights and yields, the Ravaz Index (vine yield to total pruning weight) showed the lowest value in LR treatment (1.98).

### 3.2. Effects of UV-B Exposure/Exclusion on Chemical Composition of Pinot Noir Fruit

#### 3.2.1. Amino Acids

In the 2015–2016 and 2016–2017 vintages, Pinot noir berries from the vineyard trials were taken at harvest for amino acid analysis by HPLC. Table 5 illustrated the results of the amino acid concentrations, and the percentages of the total concentration in 2015–2016. There were no significant differences between treatments for most of the amino acids on a concentration basis, except for proline (Pro). The different treatments had a prominent influence on the concentration of Pro. In the UV-B exposure treatment (LR), the fruit fully exposed to UV-B had the highest Pro concentration, at 2474 µM, whereas the shading treatment (SC) had the lowest, at 1827 µM, in grape juice. Pro in the UV-B exclusion treatment (PETG) was intermediate, at 2222 µM. This result indicated that UV-B exposure can increase Pro in berries. The percentages of histidine (His), threonine (Thr) and Pro were significantly affected by UV-B exposure and exclusion. Compared to SC (11.2%), LR caused an increase in Pro, to 16.1%. With respect to SC, no consistent significant effects from the LR and PETG treatments were found in the percentages of His and Thr. His was increased by PETG and reduced by LR, while both LR and PETG treatments caused increases in Thr. 

In 2016–2017, there were no statistically significant effects on amino acids (Table 6). Compared to SC, LR significantly increased Tryptophan (Trp), Tyrosine (Tyr), Serine (Ser), Glycine (Gly), Leucine (Leu) and Isoleucine (Ile) and decreased Valine (Val). In the aromatic family, the highest concentration of Trp and Tyr of LR, at 155 and 77 µM, respectively. UV-B exclusion treatments (PETG) caused significant reduction in Val (272 µM of PETG vs. 776 µM of PETG). LR and PETG had statistically significant effect on the percentages of individual amino acids in comparation with SC. Percentages of histidine (His), Tyr, lysine (Lys), Ser and Gly were increased by UV-B exposure (LR), compared with SC, while Val and aspartic acid (Asp) were decreased. PETG caused increases in the percentages of His (1.2%) and Asp (2%) and decreases in the percentages of Val (2.2%) and alanine (Ala) (13.1%).

#### 3.2.2. Phenolic Composition

To determine the effects of UV-B exposure and exclusion on Pinot noir fruit, samples were taken during ripening from both vintages and analyzed for skin total phenolics, skin anthocyanins, skin tannins and seed tannins. For the 2015–2016 vintage, skin anthocyanins on a per berry basis were shown in Figure 2. Values for the shading treatment (SC) remained relatively stable during ripening and peaked at 4 weeks post-veraison. The UV-B exposure treatment (LR) and UV-B exclusion (PETG) had similar patterns with SC. Compared with SC, skin anthocyanin contents were significantly increased by the UV-B exposure treatment (LR) and UV-B exclusion (PETG) from 2 weeks to 5 weeks post-veraison (harvest). At harvest, LR and PETG had 0.561 and 0.422 mg/berry, respectively, compared to 0.374 mg/berry in SC. 

The contents of skin total phenolics in SC, PETG and LR increased to the peaks at 4 weeks post-veraison and then declined (Figure 2). LR had higher skin total phenolics contents than SC from 2 weeks to 5 weeks post-veraison. Once exposed, skin total phenolics values for UV-B (LR) fruit were higher compared to those under PETG, with a rise from 3 weeks to 5 weeks post-veraison. The skin total phenolics contents were significantly higher in LR (0.334 au/berry), compared with PETG (0.252 au/berry) and SC (0.254 au/berry) at harvest. These results indicated that the major increases in skin total phenolics and anthocyanins were because of the effect of UV-B. 

During ripening, the contents of skin tannin in the treatments presented net decreases (Figure 2). SC caused decreases in skin tannin contents from 1 weeks to 4 weeks post-veraison and then increased at harvest. Skin tannins in LR went down after one week of the treatment and up in the following two weeks and decreased at harvest. Skin tannin contents in PETG increased to the peak at 2 weeks post-veraison and declined until harvest. The reduction in skin tannins for SC was 0.358 mg/berry from 1 week to 5 weeks post-veraison (harvest), while the losses for LR and PETG were 0.226 mg/berry and 0.300 mg/berry, respectively. 

During development, seed tannins also illustrated increases and then decreases in the treatments (Figure 2). The peaks of the seed tannin content in treatments were shown at 2 weeks post-veraison. PETG had the highest contents of seed tannin than SC and LR from 2 weeks to 4 weeks post-veraison, but the effects on seed tannins swapped at harvest. The contents of seed tannins in PETG were the lowest compared to SC and LR. The net increase in seed tannins at 5 weeks post-veraison increased by 73% since 1 week post-veraison in SC. With respect to SC, LR and PETG significantly decreased the contents of seed tannin with 8.450 mg/berry and 7.694 mg/berry, respectively, at 5 weeks post-veraison. 

In the 2016–2017 vintage, the skin anthocyanin contents in all treatments significantly increased from 1 week to 5 weeks post-veraison and peaked at 4 weeks post-veraison (Figure 3). In comparison to the shading treatment (SD), the UV-B exposure treatments (LR) and UV-B exclusion (PETG) had significantly higher contents of skin anthocyanin at the last two weeks of development. At harvest, LR and PETG had increased skin anthocyanins to 0.446 mg/berry and 0.242 mg/berry, respectively. The contents of skin total phenolics from 1 week post-veraison to 5 weeks post-veraison in the treatments were variable (Figure 3b). At harvest, consistent UV-B exposure responses were observed in skin total phenolics. The UV-B exposure treatment (LR) caused significant increases in skin total phenolics (0.292 au/berry), whereas skin total phenolics were reduced by PETG (0.199 au/berry), compared with the shading treatments (SC). 

In Figure 3, the contents of skin tannin increased and then reduced in all treatments from 1 weeks post-veraison to 5 weeks post-veraison. There were no significant differences between different light environmental treatments from 1 week to 3 weeks post-veraison. At harvest, PETG (0.540 mg/berry) significantly increased the contents of skin tannins and reduced skin tannins in LR (0.414 mg/berry), compared to SC (0.524 mg/berry). 

In Figure 3, the values for seed tannins fluctuated from 1 week to 5 weeks post-veraison. There were no significant effects of UV on seed tannin contents during the berry developmental stages.

#### 3.2.3. Volatile Composition

In the 2015–2016 vintage, there were no statistically significant effects from UV-B exposure and exclusion on volatile compounds in Pinot noir juice (Table 7). In the following vinetage, the treatments did not significantly affect the C6 aldehyde family at harvest. At the harvest sampling in the UV-B exposure (LR) and UV-B exclusion (PETG) treatments, (*Z*)-3-hexenol concentrations were 45.1 µg/L and 76.6 µg/L, respectively, compared with 89.5 µg/L in SC. Significant decreases in (*E*)-2-hexenol were induced by LR (11.7 µg/L) in Pinot noir juice. PETG caused significant decreases in α-terpineol and nerol, and LR increased α-terpineol, but reduced nerol at harvest, compared with SC.

## 4. Discussion

### 4.1. Alteration of Vine Yield and Pruning Weight as Induced by UV-B Radiation

Yields and pruning weights in the two vintages were not affected by PETG, SC or LR (Table 4), but differences in the Ravaz Index were shown between the treatments in 2016–2017. The Ravaz Index (fruit yield/pruning weight) is a useful parameter for reflecting the final capacity of vines given the management practices and also in evaluating vine balance [35]. Vine balance helps to maintain productive yields, fruit quality and vine health [36]. The Ravaz Index is an effect driven by differences in yields and pruning weights. In the 2016–2017 vintage, there was 241 mm rainfall from January to April, which was significantly higher than the average value for 1977–2000 (Table 4), leading to increase the yield and pruning weight. In addition, LR had the lowest Ravaz Index than SC and PETG, due to the increases in pruning weight in LR. In the 2015–2016 and 2016–2017 vintages, there was no consistent effect of UV-B on the Ravaz Index (Table 4). This result is supported by Howell [35], who stated that there were strong annual fluctuations in weather conditions during the growing season in cool climate regions.

### 4.2. The Effects of UV-B Radiation on Amino Acids in Berries

Amino acids are a very important nitrogen source for yeast metabolism during alcoholic fermentation [20,21]. Glutamine (Gln) is biosynthesized via glutamine synthetase (GS) and the glutamate synthase (GOGAT) pathway and is the major nitrogen transport compound from leaves to berries [37,38]. Light and other radiation plays an important role in the assimilation of amino acids, but there is less known about regulation of the synthesis of various amino acids in grapevines [25,39,40,41,42]. In our study, UV-B caused little to a slight increase in amounts of amino acids in the two years vineyard trials (Table 5 and Table 6). Proline as an abundant amino acid in berries was significantly changed in the vineyard trials, which may be related to the level of berry maturity and the level of stress observed. The control treatments had lower levels of °Brix than the other treatments, and °Brix in 2015–2016 was higher than in 2016–2017. Proline evolution in berry development is confined to late ripening period, from approximately 4 weeks post-veraison [24,39,43]. With respect to the accumulation of Pro during ripening, similar results were reported by Garde-Cerdán, Gutiérrez-Gamboa [44], which showed significant Pro accumulation in grapes from post-veraison and the peak of proline concentrations at 25 °Brix. Therefore, the lower level of Pro in grape juice could have been driven by the lower °Brix. Additionally, Pro in plant tissues mostly accumulates in response to osmotic stress [45]. At the late stages of ripening, the berry pulp could be affected by an increase in osmotic pressure with the increasing sugar concentration. Thus, it is possible that the accumulation of Pro occurs in response to this developmentally imposed osmotic stress and plays an osmotic-protective role in the developing berry cells [24,43]. In addition, in grapevine vegetative and reproductive tissues, ornithine can be transferred into Δ1-pyrroline-5-carboxylate (P5C) via the activity of δ-ornithine aminotransferase (OAT); and P5C is a precursor of proline [24]. However, no evidence suggested that OAT under UV-B played a regulatory role in determining the Pro concentration in berries.

### 4.3. The Phenolic Composition in Berries in Response to UV-B Radiation

The contents of skin anthocyanin and skin total phenolics showed developmental accumulation in Pinot noir berries from veraison to harvest in the vineyard (Figure 2 and Figure 3). In two vintages, compared with the shading treatment (SC), the light exposure treatment (LR) and UV-B exclusion treatment (PETG) caused an increase in skin anthocyanins and skin total phenolics during berry development, in particular, LR. These results were consistent with Del-Castillo-Alonso, Diago [46], Carbonell-Bejerano, Diago [47], Cortell and Kennedy [48] and Pastore, Zenoni [49]. It was indicated that UV-B radiation affected the biosynthesis of skin anthocyanins and skin total phenolics, due to the upregulation of the genes encoding for flavonoid 3′,5′-hydroxylase (F3′5′H) and flavonoid glucosyltransferase (UFGT) by UV-B [50,51]. Moreover, UV-B stimulates phenolic compounds in Pinot noir berry skins, which play a role as UV protectants/antioxidants [52]. The 2016–2017 vintage was a challenging season in terms of rainfall quantity, resulting in low °Brix further than the decreases in fruit phenolic composition. It could be also explained that biosynthesis of anthocyanins appeared to highly depends on berry sugar levels, of which sugar levels are not only carbohydrate sources, but also involved in the stimulation of gene activities [53,54,55]. Sucrose can upregulate F3H expression, coinciding with the enhancement of anthocyanin and phenolic composition levels [54,56,57].

Tannins are polymeric structures of flavan-3-ols and can be composed of chains of almost identical subunits. On average, skin tannins are longer than seed tannins, but the amount of seed tannins is higher than skin tannins in grapes [22,58]. In this study, the accumulation of skin tannins was significantly higher in LR at harvest, and the amount of skin and seed tannins during berry development fluctuated (Figure 2 and Figure 3). These results suggested that skin tannins may have accumulated to a greater extent under UV radiation to protect grape skins, which could be attributed to the stimulation of ANR and LAR gene expression to regulate the synthesis of skin tannins [40,50,57,59,60]. The fluctuation of skin tannins during ripening may relate to the polymeric flavan-3-ols. This polymerization is dramatically influenced by environmental factors, such as UV-B, temperature and rainfall, and changed at different stages of berry development [48,58,61]. Thus, in the 2016–2017 vintage, environmental factors near harvest may influence on the polymerization of tannins in berry skins, resulting in inconsistent results with two vintages vineyard trials. It could be because only a small part of UV-B can pass the anticlinal cell walls to impact on subdermal tissues and, as UV-B cannot penetrate far into the berry, direct impact on seed tannins is unlikely [62].

### 4.4. Effects of UV-B Radiation on Volatile Composition in Berry Juice

The C6 aldehydes, C6 alcohols and monoterpenes are the most important volatile compounds responsible for varietal aromas of grape and wine [59]. C6 compounds were not different in either year of this study, except for a reduction in (*Z*)-3-hexenol in the 2016–2017 trial, under the UV-B treatments (Table 7). This suggests that the formation and degradation of C6 compounds in berries in response to UV-B is not clear. Some alterations of C6 compounds may be because UV-B induces the catabolism of fatty acids in cell membranes [57,63]. Additionally, hexanal can be converted into other volatile compounds to play a role in defense signaling [43,64]. There was no difference from UV-B on monoterpenes in either 2015–2016 or 2016–2017, except for α-terpineol and nerol in 2016–2017. These results may align with the idea that the biosynthesis of monoterpenes in berry skins is via the 1-deoxy-D-xylulose 5-phosphate/2-C-methyl-D-erythritol 4-phosphate (DOXP/MEP) pathway in plastids and the mevalonate pathway in cytosols. In these pathways, phenyl diphosphates are catalyzed by terpene synthases (TPS) to produce an abundance of terpenes. UV-B stimulates TPS, leading to the synthesis of monoterpenes [65,66,67]. Moreover, UV-B stimulated the production of ROS, which caused the generation of monoterpenes to protect grapes from oxidative damage [68,69,70]. The stimulation of monoterpenoids by UV enhanced the strong aroma and contributed to the characteristic fragrances (e.g., rose, lilac, pine and citrus) in grapes, which could be transferred into final wine [22,71]. In this paper, the free-form aromas have been investigated, but there are also many glycosidically bound aroma compounds in grapes, which can be released by enzymatic and acidic hydrolysis during crushing, storage and fermentation [17]. Thus, it is important for future work to understand how the environmental issues affect the glycosidically bound aromas.

## 5. Conclusions

This research clearly demonstrated that the physiology of *Vitis vinifera* L. var. Pinot noir vines was altered by UV-B. Sunlight exposure with or without UV-B does not affect fruit production capacity and °Brix in the vineyard. Berries exposed to sunlight with or without UV-B resulted in no changes in total amino acids over the two years. This research reports increased contents of skin anthocyanin and skin total phenolics in fruit under UV-B. When berries were directly exposed to UV-B, more phenolic compounds accumulated in berry skins. The research showed that there were no changes in the amount of C6 compounds of Pinot noir juice from the vineyard. The concentration of nerol was increased by UV-B in the vineyard in 2016–2017. The information presented also indicated how important leaf canopy management was for vineyard management. Therefore, UV radiation potentially affected fruit ripeness and crop load, and even stimulated the accumulation of phenolic compounds that may affect Pinot noir quality. This research substantially contributed to improving our scientific understanding of UV-B responses in an important commercial species. In addition, it provided valuable information for possible changes to vineyard management and highlighted some future research to produce high-quality wines with the anticipated specific characteristics.

## Figures and Tables

**Figure 1 foods-12-02350-f001:**
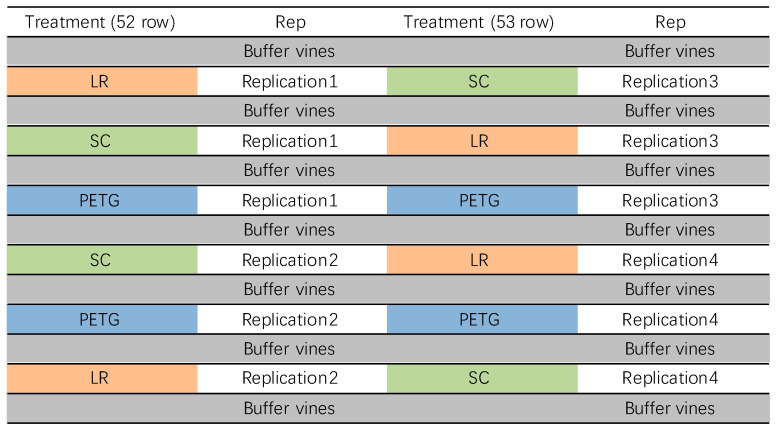
Vineyard experimental design. Data showed the trial design of this study. SC (green): shade cloth treatment; LR (yellow): leaf removal treatment; PETG (blue): polyethylene terephthalate screen treatment.

**Figure 2 foods-12-02350-f002:**
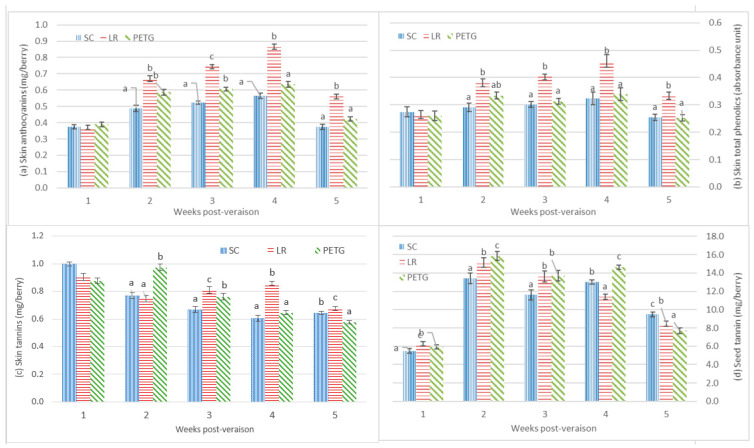
Effects of UV-B exposure/exclusion on skin anthocyanins, skin total phenolic substances, skin tannins and seed tannins in Pinot noir berries during ripening in the 2015–2016 vineyard trials. Data showed the mean ± standard error of four replicates. *p*-values for statistical significance compared the different treatments according to one-way ANOVA and LSD test at the 5% level. Different letters indicate significant differences at *p* < 0.05.

**Figure 3 foods-12-02350-f003:**
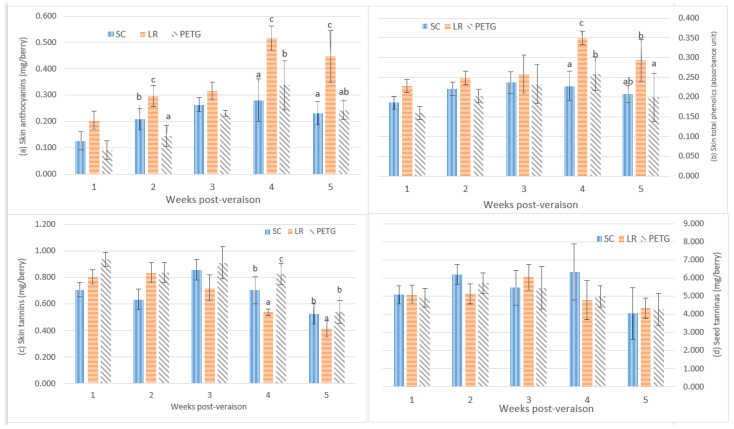
Effects of UV-B exposure/exclusion on skin anthocyanins, skin total phenolic substances, skin tannins and seed tannins in Pinot noir berries during ripening in the 2016–2017 vineyard trials. Data showed the mean ± standard error of four replicates. p-values for statistical significance compared the different treatments according to one-way ANOVA and LSD test at the 5% level. Different letters indicate significant differences at *p* < 0.05.

**Table 1 foods-12-02350-t001:** Volumes of samples and reagents for MCP tannin assay for grape extractions.

	Sample (mL)	MCP (mL)	(NH_4_)_2_SO_4_ (mL)	Water (mL)
Treatment	0.1	0.3	0.2	0.4
Control	0.1	0	0.2	0.7

**Table 2 foods-12-02350-t002:** Deuterated and non-deuterated standards for six C_6_ and five monoterpene volatile compounds in Pinot noir juice.

Compound	ISTD ID No	RT (min)	Target Ion (*m*/*z*)	Qualifier Ions (*m*/*z*, % of Target)	Calibration Range ^ (µg/L)	CAS No.
d_12_ Hexanal	ISTD 1	7.78	64	48 (140.2), 46 (92.6)	-	1219803-74-3
n-Hexyl d_13_ Alcohol	ISTD 2	10.12	64	50 (45.2), 46 (44.1)	-	16416-34-5
d_3_ Linalool	ISTD 3	12.31	96	124 (25.9), 139 (10.1), 58 (16.8)	-	1216673-02-7
Hexanal	1	7.85	44	41 (77.8), 56 (75.2)	0–1048.6	66-25-1
(*E*)-2-Hexenal	2	9.23	41	55 (74.4), 39 (59.5)	0–1517.1	6728-26-3
1-Hexanol	2	10.26	56	43 (64.5), 55 (51.3)	0–824.1	111-27-3
(*E*)-3-Hexen-1-ol	2	10.33	67	82 (58.1), 100 (3.8)	0–23.4	928-97-2
(*Z*)-3-Hexen-1-ol	2	10.53	41	67 (78.2), 55 (38.8)	0–265.4	928-96-1
(*E*)-2-Hexen-1-ol	2	10.70	57	41 (50), 39 (20.5)	0–513.3	928-95-0
Linalool	3	12.35	93	12 (28.0), 136 (8.8)	0–8.6	78-70-6
Citronellol	3	15.44	138	82 (468.2), 95 (397.3), 109 (138.2)	0–8.2	7540-51-4
α-terpineol	3	14.59	93	121 (75.8), 136 (60.9), 81 (61.36)	0–6.3	10482-56-1
Nerol	3	16.06	68	123 (28.9), 139 (18.1), 136 (11.4)	0–7.3	106-25-2
Geraniol	3	16.88	84	93 (122.3), 123 (98.9)	0–13.3	106-24-1

^ All samples were diluted 2-fold with 0.2 mol/L citrate buffer, hence concentrations obtained were multiplied by this factor accordingly.

**Table 3 foods-12-02350-t003:** Effects of UV-B exposure and exclusion on berry parameters, yield, pruning weight and the Ravaz Index in Pinot noir at harvest in the 2015–2016 and 2016–2017 vineyard trials.

Vintage	Treatment	°Brix	TA (g/L)	pH	Vine Yield (kg)	Pruning Weight (kg)	Ravaz Index
2015–2016	SC	^a^ 20.5	8.6	3.53	2.58	0.82	3.38
	LR	^b^ 21.6	8.6	3.58	2.47	0.66	4.28
	PETG	^b^ 21.6	8.2	3.59	2.31	0.72	3.32
	*p*-value	0.028	n.s	n.s	n.s	n.s	n.s
2016–2017	SC	17.1	^b^ 11.4	3.65	3.43	0.62	^b^ 5.55
	LR	18.2	^a^ 10.4	3.76	1.60	1.21	^a^ 1.98
	PETG	16.4	^a^ 10.5	3.62	2.97	0.71	^b^ 4.23
	*p*-value	n.s	0.036	n.s	n.s	n.s	0.020

Data showed the mean of four replicates from samples at harvest in 2015–2016 and 2016–2017. *p*-value, significance of light exposure effect according to one-way ANOVA and LSD test at the 5% level. Different letters indicate significant differences at *p* < 0.05; n.s, no significance; PETG, PETG screen; LR, UV-B exposure; SC, shade cloth.

**Table 4 foods-12-02350-t004:** Monthly rainfall and solar irradiance of the vineyard in 2015/2016 and 2016/2017.

		Rainfall (mm)	Rad (MJm^2^)	Temperature max (°C)	Temperature min (°C)	Average Temperature (°C)
Average values for 1971–2000	January	42	678.9	22.6	11.4	17
	February	39	526.4	21.7	11	16.3
	March	54	437.1	20.1	9.9	15
	April	54	291	17.5	6.7	12.2
	Total	189	1933.4	/	/	/
2016	January	107	578.52	22.6	11.4	17
	February	24	600.15	21.7	11	16.3
	March	34	460.04	20.1	9.9	15
	April	10	325.74	17.5	6.7	12.2
	Total	175	1964	/	/	/
2017	January	42	705.5	23.2	11.3	17.2
	February	3	550.4	23	11.4	17.2
	March	73	380.2	19.2	10.6	14.9
	April	123	260	16.5	8.1	12.3
	Total	241	1896.1	/	/	/

**Table 5 foods-12-02350-t005:** Effects of UV-B exposure/exclusion on amino acids and the percentages of each amino acid in total amino acids in Pinot noir berries at harvest in the 2015–2016 vineyard trials.

Amino Acid (µM)	Treatment			*p*-Value	Treatment			*p*-Value
	SC	LR	PETG		SC	LR	PETG	
*α-ketoglutarate*								
Pro	^a^ 1827	^b^ 2474	^ab^ 2222	0.025	^a^ 11.2%	^b^ 16.1%	^ab^ 15.6%	0.007
Arg	6225	5949	5444	n.s	38.2%	39.5%	38.1%	n.s
Glu	246	211	222	n.s	1.5%	1.4%	1.6%	n.s
Gln	1855	856	866	n.s	11.4%	5.7%	6.1%	n.s
His	283	217	253	n.s	^b^ 1.7%	^a^ 1.4%	^b^ 1.8%	0.011
*Shikimate* *(aromatic)*								
Phe	443	341	315	n.s	2.7%	2.3%	2.2%	n.s
Trp	144	121	123	n.s	0.9%	0.8%	0.9%	n.s
Tyr	29	20	22	n.s	0.2%	0.1%	0.2%	n.s
*Pyruvate*								
Leu	515	444	451	n.s	3.2%	2.9%	3.2%	n.s
Val	332	289	284	n.s	2.0%	1.9%	2.0%	n.s
Ala	1647	1489	1482	n.s	10.1%	9.9%	10.4%	n.s
*Aspartate*								
Asp	281	261	252	n.s	1.7%	1.7%	1.8%	n.s
Asn	101	52	67	n.s	0.6%	0.3%	0.5%	n.s
Thr	1157	1166	1153	n.s	^a^ 7.1%	^ab^ 7.7%	^b^ 8.1%	0.033
Ile	368	309	301	n.s	2.3%	2.1%	2.1%	n.s
Met	89	73	71	n.s	0.5%	0.5%	0.5%	n.s
Lys	66	64	63	n.s	0.4%	0.4%	0.4%	n.s
*3-phosphoglycerate*								
Cys	N.A	N.A	N.A	N.A	N.A	N.A	N.A	N.A
Ser	656	669	661	n.s	4.0%	4.4%	4.6%	n.s
Gly	29	32	35	n.s	0.2%	0.2%	0.2%	n.s
Total	16,305	15,052	14,282	n.s				

Data showed the mean of four replicates from samples at harvest in 2015–2016. *p*-values for statistical significance comparing the different treatments according to one-way ANOVA and LSD test at the 5% level. Different letters indicate significant differences at *p* < 0.05; n.s, no significance; N.A, not available.

**Table 6 foods-12-02350-t006:** Effects of UV-B exposure/exclusion on amino acids and the percentages of each amino acid in total amino acids in Pinot noir berries at harvest in the 2016–2017 vineyard trials.

Amino Acid (µM)	Treatment			*p*-Value	Treatment			*p*-Value
	SC	LR	PETG		SC	LR	PETG	
*α-ketoglutarate*								
Pro	516	1332	748	n.s	4.1%	8.1%	6.1%	n.s
Arg	3256	2854	3349	n.s	25.8%	17.4%	27.2%	n.s
Glu	276	389	262	n.s	2.2%	2.4%	2.1%	n.s
Gln	2180	3419	2261	n.s	17.2%	20.8%	18.4%	n.s
His	117	192	148	n.s	^a^ 0.9%	^b^ 1.2%	^b^ 1.2%	0.009
*Shikimate (aromatic)*								
Phe	477	582	465	n.s	3.8%	3.5%	3.8%	n.s
Trp	^a^ 69	^b^ 155	^ab^ 114	0.033	0.5%	0.9%	0.9%	n.s
Tyr	^a^ 52	^b^ 77	^a^ 52	0.036	^a^ 0.4%	^b^ 0.5%	^a^ 0.4%	0.045
*Pyruvate*								
Leu	^a^ 397	^b^ 577	^a^ 372	0.032	3.1%	3.5%	3.0%	n.s
Val	^c^ 776	^b^ 526	^a^ 272	0.006	^c^ 6.1%	^b^ 3.2%	^a^ 2.2%	<0.001
Ala	1895	2749	1612	n.s	^b^ 15.0%	^b^ 16.7%	^a^ 13.1%	0.023
*Aspartate*								
Asp	218	242	242	n.s	^ab^ 1.7%	^a^ 1.5%	^b^ 2.0%	0.005
Asn	67	97	65	n.s	0.5%	0.6%	0.5%	n.s
Thr	1061	1375	1199	n.s	8.4%	8.4%	9.7%	n.s
Ile	^a^ 309	^b^ 460	^a^ 304	0.029	2.4%	2.8%	2.5%	n.s
Met	169	173	40	n.s	1.3%	1.1%	0.3%	n.s
Lys	31	50	40	n.s	^a^ 0.2%	^b^ 0.3%	^b^ 0.3%	0.042
*3-phosphoglycerate*								
Cys	6	10	12	n.s	0.1%	0.1%	0.1%	n.s
Ser	^a^ 741	^b^ 1113	^a^ 728	0.030	^a^ 5.9%	^b^ 6.8%	^a^ 5.9%	0.015
Gly	^a^ 30	^b^ 56	^a^ 28	0.033	^a^ 0.2%	^b^ 0.3%	^a^ 0.2%	0.026
Total	12,646	16,429	12,314	n.s				

Data showed the mean of four replicates from samples at harvest in 2016–2017. *p*-values for statistical significance comparing the different treatments according to one-way ANOVA and LSD test at the 5% level. Different letters indicate significant differences at *p* < 0.05; n.s, no significance.

**Table 7 foods-12-02350-t007:** Effects of UV-B exposure/exclusion on volatile compounds in Pinot noir juice at harvest in the 2015–2016 and 2016–2017 vineyard trials.

Vintage	Treatments	C_6_ Aldehydes (µg/L)		C_6_ Alcohols (µg/L)				Free Monoterpenes (µg/L)				
		Hexanal	(*E*)-2-Hexenal	Hexanol	(*E*)-3-Hexenol	(*Z*)-3-Hexenol	(*E*)-2-Hexenol	Linalool	*α*-Terpineol	Citronellol	Nerol	Geraniol
2015–2016	SC	34.7	66.4	749.7	11.9	38.4	401.3	1.7	1.6	1.2	4.7	17.6
	LR	44.5	67.8	671.0	11.7	43.1	404.5	1.7	1.4	1.0	4.8	16.9
	PETG	33.8	71.1	754.3	11.6	41.5	446.2	1.7	1.6	1.0	5.1	18.2
	*p*-value	n.s	n.s	n.s	n.s	n.s	n.s	n.s	n.s	n.s	n.s	n.s
2016–2017	SC	35.0	35.1	1118.2	17.4	^b^ 76.6	^b^ 148.9	1.8	^b^ 1.4	1.0	^a^ 4.8	12.7
	LR	43.7	51.9	1064.8	15.7	^a^ 45.1	^a^ 11.7	1.8	^b^ 1.3	1.2	^b^ 8.5	13.3
	PETG	31.4	33.9	1062.2	14.9	^b^ 89.5	^b^ 156.1	1.7	^a^ 1.1	1.0	^a^ 4.4	13.8
	*p*-value	n.s	n.s	n.s	n.s	0.042	<0.001	n.s	0.012	n.s	<0.001	n.s

Data showed the mean of four replicates from samples at harvest in 2015–2016 and 2016–2017. *p*-value, significance of light exposure effect according to one-way ANOVA and Fisher’s LSD test at the 5% level. Different letters indicate significant differences at *p* < 0.05; n.s, no significance; PETG, PETG screen; LR, UV-B exposure; SC, shade cloth.

## Data Availability

Data is contained within the article.

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
