# Peer review of "UV-B Radiation Induced the Changes in the Amount of Amino Acids, Phenolics and Aroma Compounds in Vitis vinifera cv. Pinot Noir Berry under Field Conditions"

_foods, 2023, doi:10.3390/foods12122350_

Round 1

Reviewer 1 Report

The manuscript evaluated the impact of UV-B radiation on amino acids, phenolics and aroma composition Vitis vinifera cv. Pinot. The research is well designed, the results are statistically analyzed and well discussed. Some important point need to be reviewed:

·       A strong English revision is necessary

·       Table 1: please change it to a figure and add R1-4 meaning in the legend

·       All equations must be formatted, numbered and called in the text.

·       Item 2.4.2: please add reference for samples preparation

·       Figure 1: please add equation and R2 value to the figure.

·       Tables are not in the right order of appearance. Table 8 appears before table 5. Please set Tables the nearest of their citation in the text.

·       Table 5 font is not in journal standard and please add Temperature unit

Author Response

We thank the reviewers and editor for their assessment of the manuscript. Their valuable comments and suggestions have helped improve the quality of our study. Below we addressed all comments through a point-by-point response.

1) A strong English revision is necessary.

We thank the reviewer for this comment. We asked someone else to check the grammar of the main text and did the modification.

2) Table 1: please change it to a figure and add R1-4 meaning in the legend

Thanks for this suggestion. We changed Table 1 to figure and added the meaning of R1-4.

3) All equations must be formatted, numbered and called in the text.

We thank the reviewer for this suggestion, We modified the format of equations.

4) Item 2.4.2: please add reference for samples preparation.

Thanks for this suggestion, we added the reference for samples preparation.

5) Figure 1: please add equation and R2 value to the figure.

We thank for this suggestion, We added the the equation and R2 of the calibration curve.

6) Tables are not in the right order of appearance. Table 8 appears before table 5. Please set Tables the nearest of their citation in the text.

Thanks for this suggestions, we modified the order of tables and figure, and insert them after the text where first mentioned.

7) Table 5 font is not in journal standard and please add Temperature unit

Thanks for this suggestion, we add the temperature unit.

Reviewer 2 Report

The manuscript addresses a valuable topic and raises interesting questions. However, some major revisions are needed

        Table 4 and Table 8. Reported: Data showed mean ± standard error. Reality: Missing Standard Errors

        The formula in Line 143 must be formated

        The calibration curve (Figure 1 Epicatechin equivalent calibration curve) in Line 193 is superfluous.  Instead add equation.

        The discussion is well-written but lengthy. It could be shortened. The authors failed to discuss the weaknesses of the present study.

·        The results are difficult to follow. I suggest to split the Figure 2 and Figure 3 into several images. And insert Figures after text, where first mentioned.

Author Response

We thank the reviewers and editor for their assessment of the manuscript. Their valuable comments and suggestions have helped improve the quality of our study. Below we addressed all comments through a point-by-point response.

1)Table 4 and Table 8. Reported: Data showed mean ± standard error. Reality: Missing Standard Errors

Thanks for this suggestion, we indicated that the data showed the mean of four replicates from samples at harvest.

2) The formula in Line 143 must be formated.

We thank the reviewer for this suggestion, We modified the format of equations.

3) The calibration curve (Figure 1 Epicatechin equivalent calibration curve) in Line 193 is superfluous.  Instead add equation.

We thanks for this suggestion, We added the the equation of the calibration curve.

4) The discussion is well-written but lengthy. It could be shortened. The authors failed to discuss the weaknesses of the present study.

Thanks for this comment, we discussed the strong annual fluctuations in weather have effects on our results and we will continue our research in next few years to obtain more data.  

5) The results are difficult to follow. I suggest to split the Figure 2 and Figure 3 into several images. And insert Figures after text, where first mentioned.

Thanks for this suggestions, we modified the order of tables and figure, and insert them after the text where first mentioned.

Reviewer 3 Report

The paper entitled »UV-B radiation-induced the changes of amino acids, phenolics and aroma compounds in Vitis vinifera cv. Pinot noir berry under field conditions« of Sun et al is reporting the influence of sunlight exposure (more specifically: of UV-B radiation) on the content of skin anthocyanin and skin total phenolics in the berry of Pinot noir grape variety of the species Vitis vinifera. The field study lasted two years and authors concluded that increased UV-B radiation leads to increased content of skin anthocyanin and skin total phenolics. The results for amino acids were not so uniform and were dependent on the kind of the amino acid, nevertheless, the influence of changed UV-B radiation on production of amino acids was in general small, one may perhaps even say marginal.

The paper is in general written in an understandable manner and grammatically correctly, although some passages require very attentive reading due to many details given in a condensed manner.

The paper might be interesting for wine producers and in my opinion the research is done correctly enough to satisfy criteria in Foods journal after minor revision is done.

My remarks:

1.)    The title of the paper is perhaps a bit misleading. The notation »UV-B radiation-induced the changes of amino acids, phenolics, and aroma compounds …” may be understood that UV-B radiation chemically changes amino acids, phenolics, and aroma compounds (through stimulating some chemical reactions leading to transformation of amino acids, ….  to some kind of their derivatives). The study is not about the chemical changes, the study is about the changes of the amount (or the content) of amino acids, phenolics, and aroma compounds in Vitis vinifera. My suggestion to authors is to consider this small modification of the title of the manuscript.

2.)    Due to many variables that may influence the growth of plants (in this case Vitis vinifera cv. Pinot noir) it would be good to monitor change of the content of the compounds of the interest through the longer period (perhaps five years), not just two years, in order to obtain more reliable results. On the other hand, I can also understand, that nowadays carrying out such a study for a longer time is difficult due to pressure to publish.

3.)    Some abbreviations and less know terms appear in the text without being explained immediately when they appear in the text for the first time (e.g. C6, PETG, Ravaz index). My suggestion to authors is to explain such abbreviations/terms immediately at the place where they appear for the first time.

4.)    I suggest that the sentence “The research showed there were no changes in C6 compounds in the vineyard.” (Line 490) is changed. I assume that the authors probably wanted to say that there were no changes of the amount/content of C6 in Pinot noir juice.

5.)    Size and placement of equations (lines 143, 179, 183, 214) is annoying.

6.)    Figure 1 and Tables 1, 4 and 5 are not aligned with the text column.

7.)    Numbers on graph ordinates are given to more places than needed (Figures 1, 2d, 3).

English in is general fine, my minor suggestions for improving the language are included in my review

Author Response

We thank the reviewers and editor for their assessment of the manuscript. Their valuable comments and suggestions have helped improve the quality of our study. Below we addressed all comments through a point-by-point response.

1) The title of the paper is perhaps a bit misleading. The notation »UV-B radiation-induced the changes of amino acids, phenolics, and aroma compounds …” may be understood that UV-B radiation chemically changes amino acids, phenolics, and aroma compounds (through stimulating some chemical reactions leading to transformation of amino acids, ….  to some kind of their derivatives). The study is not about the chemical changes, the study is about the changes of the amount (or the content) of amino acids, phenolics, and aroma compounds in Vitis vinifera. My suggestion to authors is to consider this small modification of the title of the manuscript.

We thank the reviewer for this comment. We modified the title of this manuscript : UV-B radiation-induced the changes of the amounts of amino acids, phenolics and aroma compounds in Vitis vinifera cv. Pinot noir berry under field conditions

2) Due to many variables that may influence the growth of plants (in this case Vitis vinifera cv. Pinot noir) it would be good to monitor change of the content of the compounds of the interest through the longer period (perhaps five years), not just two years, in order to obtain more reliable results. On the other hand, I can also understand, that nowadays carrying out such a study for a longer time is difficult due to pressure to publish.

We thank the reviewer for this suggestion. We will continue our research in the next few years to obtain more data.

3) Some abbreviations and less know terms appear in the text without being explained immediately when they appear in the text for the first time (e.g. C6, PETG, Ravaz index). My suggestion to authors is to explain such abbreviations/terms immediately at the place where they appear for the first time.

Thanks for this suggestion. We added the explanation of the abbreviations at the place where they appear for the first time.

4) I suggest that the sentence “The research showed there were no changes in C6 compounds in the vineyard.” (Line 490) is changed. I assume that the authors probably wanted to say that there were no changes of the amount/content of C6 in Pinot noir juice.

Thanks for this suggestion. We revised this sentence according to the suggestions.

5) 6) 7) Size and placement of equations (lines 143, 179, 183, 214) is annoying.Figure 1 and Tables 1, 4 and 5 are not aligned with the text column. Numbers on graph ordinates are given to more places than needed

Thanks for this suggestion. We modified the format of equations, tables and figures.